# Utilizing Edible Agar as a Carrier for Dual Functional Doxorubicin-Fe_3_O_4_ Nanotherapy Drugs

**DOI:** 10.3390/ma14081824

**Published:** 2021-04-07

**Authors:** Yu-Jyuan Wang, Pei-Ying Lin, Shu-Ling Hsieh, Rajendranath Kirankumar, Hsin-Yi Lin, Jia-Huei Li, Ya-Ting Chen, Hao-Ming Wu, Shuchen Hsieh

**Affiliations:** 1Department of Nursing, Kaohsiung Armed Forces General Hospital, 2 Zhongzheng 1st Rd., Kaohsiung 80284, Taiwan; jyuan0418@gmail.com; 2Department of Chemistry, National Sun Yat-sen University, 70 Lien-Hai Rd., Kaohsiung 80424, Taiwan; phoebe00315@yahoo.com.tw (P.-Y.L.); r7.kirankumar@gmail.com (R.K.); xinyi062440@gmail.com (H.-Y.L.); 3Department of Seafood Science, National Kaohsiung University of Science and Technology, 142 Haijhuan Rd., Kaohsiung 81157, Taiwan; slhsieh@ntu.edu.tw (S.-L.H.); s24261102@gmail.com (J.-H.L.); 4College of Hydrosphere Science, National Kaohsiung University of Science and Technology, 142 Haijhuan Rd., Kaohsiung 81157, Taiwan; melodyyu.chen@gmail.com; 5Division of Cardiology, Department of Internal Medicine, Kaohsiung Armed Forces General Hospital, 2 Zhongzheng 1st Rd., Kaohsiung 80284, Taiwan; motherbbs@gmail.com; 6School of Pharmacy, College of Pharmacy, Kaohsiung Medical University, 100 Shih-Chuan 1st Rd., Kaohsiung 80708, Taiwan; 7Regenerative Medicine and Cell Therapy Research Center, Kaohsiung Medical University, 100 Shih-Chuan 1st Rd., Kaohsiung 80708, Taiwan

**Keywords:** iron oxide, doxorubicin, ferromagnetic nanoparticles, drug delivery, synergic cytotoxic effects

## Abstract

The purpose of this study was to use agar as a multifunctional encapsulating material to allow drug and ferromagnetism to be jointly delivered in one nanoparticle. We successfully encapsulated both Fe_3_O_4_ and doxorubicin (DOX) with agar as the drug carrier to obtain DOX-Fe_3_O_4_@agar. The iron oxide nanoparticles encapsulated in the carrier maintained good saturation of magnetization (41.9 emu/g) and had superparamagnetism. The heating capacity test showed that the specific absorption rate (SAR) value was 18.9 ± 0.5 W/g, indicating that the ferromagnetic nanoparticles encapsulated in the gel still maintained good heating capacity. Moreover, the magnetocaloric temperature could reach 43 °C in a short period of five minutes. In addition, DOX-Fe_3_O_4_@agar reached a maximum release rate of 85% ± 3% in 56 min under a neutral pH 7.0 to simulate the intestinal environment. We found using fluorescent microscopy that DOX entered HT-29 human colon cancer cells and reduced cell viability by 66%. When hyperthermia was induced with an auxiliary external magnetic field, cancer cells could be further killed, with a viability of only 15.4%. These results show that agar is an efficient multiple-drug carrier, and allows controlled drug release. Thus, this synergic treatment has potential application value for biopharmaceutical carrier materials.

## 1. Introduction

The design and synthesis of effective drug delivery carrier materials holds critical importance in biomedicine. However, studies have shown that conventional chemotherapy may lead to drug resistance during treatment [1,2]. Therefore, nanoparticles have been widely used as carrier delivery systems in many studies. In addition to improving drug delivery to tumor cells and prolonging drug efficacy, the use of combined chemotherapy has also been widely studied in order to increase the functionality of particle carriers [3]. Based on published research, drug delivery systems require three key features: high drug-loading efficiency [4,5], controlled drug release [6,7], and the safe decomposition of drug carriers [8]. Therefore, carrier materials from organic natural materials or inorganic synthetic products have been widely developed in recent years, such as 1D coordination polymer or carbon nanotubes [9,10], kaolinite nanoclay carrier with 2D structure [11], mesoporous organosilica with 3D structure [12,13,14], exosome [15], natural liposomes or natural polymer agar, biomacromolecular scaffolds [16], and magnetic composite, etc. Among natural polymers, agar that is extracted from red algae and which composition is galactose polymer is a recognized non-toxic substance by most countries in the world [17]. It has high drug-loading efficiency and contains micropores in a gel state [18], which allows for controlled drug release [19]. It is edible and can thus be decomposed and absorbed for biological utilization, making it an excellent low-toxicity porous material. Doxorubicin (DOX) is a common chemotherapy drug. As a DNA intercalant, it can inhibit topoisomerase II, covering DNA during gene transcription, inhibiting recombination of DNA double strands and DNA replication, and thus causing cell apoptosis [20]. Many studies are still facing challenges to reduce the adverse side effects of chemotherapy drugs, such as the short biological lifetime and dose-dependent side effects [21]. In order to improve the function of carrier particles, we used a magnetic material to induce physical hyperthermia, which can help reduce the dosage of chemotherapy drugs, thereby reducing the side effects of drugs. Iron oxide (Fe_3_O_4_) magnetic nanoparticles are the best choice. Fe_3_O_4_ is a superparamagnetic material with high saturation magnetization capacity, good biocompatibility, and low toxicity [22], and is thus widely used in biological fields. On the other hand, magnetic hyperthermia mediated by Fe_3_O_4_ nanoparticles, which allows for in vivo deep penetration, has proven to be an alternative and promising approach for cancer treatment. When subjected to an alternating magnetic field, superparamagnetic Fe_3_O_4_ nanoparticles can convert electromagnetic energy into heat through the oscillation of their magnetic moments. In this study, agar was successfully used as a drug carrier to encapsulate Fe_3_O_4_ and DOX simultaneously by means of co-precipitation to form a DOX-Fe_3_O_4_@agar nanoparticle. Then, the DOX-Fe_3_O_4_@agar carrier particles were used to treat colon cancer HT29 cells, after which the cell viability was evaluated. We also explored the release effect of the chemotherapy drug as well as the cytotoxic effect on cancer cells in a controlled magnetic-induced hyperthermia test. To do so, we induced alternating magnetic field (AMF) to generate heat from the encapsulated iron oxide magnetic particles. This allowed us to understand whether the gel carrier particles can achieve a synergistic effect, improve carrier functionality, and reduce the drug dosage.

## 2. Materials and Methods

### 2.1. Materials

Agar was purchased from Fisher chemical, Pittsburgh, PA, USA. Iron (II) chloride and iron (III) chloride were from Merck, Germany. DOX was from Sigma-Aldrich, Saint Louis, MO, USA. MTT test was from EMD Millipore, USA. Isopropanol was from TEDIA, USA. Phosphate-buffered saline (PBS) solution (pH 7) was from UniRegion Bio-Tech, Taiwan and used as obtained without further purification. Milli-Q reagent-grade water (18.2 MΩ cm at 25 °C) was used for all synthesis processing steps that required water. 

### 2.2. Synthesis of DOX-Fe_3_O_4_@agar Nanoparticles

For the co-precipitation synthesis method of Fe_3_O_4_@agar, refer to Hsieh et al. [23]. 0.05 g agar in 1 mL water was prepared, boiled to dissolve, and then cooled to form gel blocks. The gel was soaked in a mixed solution of 0.5 M iron (II) chloride and 1 M iron (III) chloride at room temperature, and the gel was washed three times with Milli-Q after reacting for 12 h. The gel was immersed again in 2.5 M sodium hydroxide for 1 h. Then, the gel was washed with ultrapure water to obtain Fe_3_O_4_@agar NPs. Finally, Fe_3_O_4_@agar NPs were heated to 38 °C, mashed, and vacuum-dried after rapidly adding DOX cooling gel. The dried gel was ground into powder using a mortar to obtain DOX-Fe_3_O_4_@agar NPs before storing them at 4 °C protected from light before the experiment.

### 2.3. Characterization

Transmission Electron Microscopy (TEM) images of Fe_3_O_4_ and DOX-Fe_3_O_4_@agar were acquired using a JEOL JEM-2100 (Tokyo, Japan) operated at 200 KV. The TEM sample were suspended in Deionized water and supported onto a carbon composite TEM grid with copper 200 mesh and allowed to dry. The average particle dimension of the Fe_3_O_4_ NPs was estimated by image J software (version Java 1.8.0). The crystalline structure was characterized using powder X-ray diffraction (XRD) with a Bruker D8 ADVANCE (Billerica, MA, USA) diffractometer equipped with Ni-filtered Cu Kα (λ = 1.5406 Å) radiation source, over a 2θ range of 20° to 80° with scan step 0.1 degrees, scan rate 3 s/step. The average particle size (*D*) of the Fe_3_O_4_ NPs was estimated by the Scherrer’s formula using the highest intensity XRD peak.
$$\mathrm{crystallite}\;\mathrm{size}\;D=\;\frac{K\times \lambda }{\beta_{2\theta }\times \cos\theta }$$
where *K* is Scherrer constant related to crystallite shape (*K* = 0.9), λ is the wavelength of the X-rays, β_2θ_ is full-width at half-maximum diffraction peak (FWHM), and θ is a diffraction angle.

Room-temperature magnetization curves for Fe_3_O_4_ and DOX-Fe_3_O_4_@agar were measured using a Quantum Design MPMSXL7 (San Diego, CA, USA) superconducting quantum interference device (SQUID). The fluorescence microscopy images of HT-29 cells emitting appropriate fluorescence were recorded with the Olympus IX73 (Tokyo, Japan) inverted fluorescence microscope equipped with the tetramethylrhodamine (TRITC) channel, at original magnification 100-fold.

### 2.4. Heating Capacity of DOX-Fe_3_O_4_@agar Nanoparticles

The DOX-Fe_3_O_4_@agar NPs (5 mg/mL) were prepared in Dulbecco’s Modified Eagle Medium (Gibco, Thermo Fisher Scientific, Inc., Waltham, MA, USA) (pH 7) by sonication. Their heating capacity was measured by portable electric induction heat machine (GE-H6, JIE HONG XING Technology. Co., LTD, Taiwan). The sample was placed at the center of the induction coil, and an alternating current of 450 A/m and a fixed frequency of 400 kHz were applied for up to 15 min. The temperature of the sample was recorded using an alcohol thermometer every 30 s. The experiments were repeated at least three times. We examined the hyperthermia effect of the DOX-Fe_3_O_4_@agar NPs on HT-29 cells. HT-29 cells were cultured in 5 mg/mL of DOX-Fe_3_O_4_@agar NPs for 24, 48, and 72 h, respectively. Then, treated HT-29 cells were exposed to the center of the induction coil for 15 min. Finally, the viability of the cells was studied by MTT assay.

### 2.5. Cell Culture

HT-29 cell line (human colon adenocarcinoma cells) was cultured in Dulbecco’s modified eagle medium (DMEM) (supplemented with 10% fetal bovine serum and 1% penicillin/streptomycin), at 37 °C in a humidified atmosphere containing 5% CO_2_ [24]. Cell viability of HT-29 was measured by the colorimetric MTT test. The cells (1 × 10^6^ cells) were seeded in 3-cm plates for 24 h and treated with 3 µg/mL, 6 µg/mL, and 12 µg/mL Fe_3_O_4_@agar or DOX-Fe_3_O_4_@agar for 24, 48, and 72 h. After the incubation periods, the cells were further incubated with MTT (0.1 mg/mL in medium) for 3 h at 37 °C to allow viable cells with active metabolism to produce a purple-colored formazan. The reaction was then stopped with 100 µL isopropanol, and the optical density (OD) was measured at 570 nm. The cell viability was expressed as a percentage of the viability of the control culture.

### 2.6. Drug Release Test

The drug release test of DOX-Fe_3_O_4_@agar in phosphate-buffered saline (PBS) solution (pH 7). In a typical experiment, 5 mg of DOX-Fe_3_O_4_@agar was added to 1 mL of PBS solution. At the given reaction time interval (record every 2 min) for 60 min, the DOX-Fe_3_O_4_@agar were separated using an external magnetic field, then the supernatant was collected for analysis of DOX concentration. After each measurement, the DOX-Fe_3_O_4_@agar sample was placed back into the release solution to keep the solution constant. The release concentration of DOX was determined using a UV–vis spectrometer (Hitachi U3900, Tokyo, Japan) at an absorption wavelength of 480 nm.

### 2.7. Statistical Analysis

Cell viability data were analyzed using the statistical analysis software SPSS for Windows, version 12.0 (SPSS Inc., Chicago, IL, USA). Data were subject to a one-way ANOVA test and where significant differences among means were found, these were separated by Duncan’s multiple range tests. OriginPro, version 8.1 (OriginLab, Northampton, MA, USA) was used for preparing figures.

## 3. Results

### 3.1. Characteristic Analysis of DOX-Fe_3_O_4_@agar

Figure 1a is a Transmission electron microscopy (TEM) image of DOX-Fe_3_O_4_@agar. The results show that DOX-Fe_3_O_4_@agar was uniformly distributed in granular form. The average particle size of encapsulated iron oxide calculated on TEM images was 9.2 ± 3.2 nm. The lattice structure of DOX-Fe_3_O_4_@agar was analyzed by X-ray diffraction (XRD), as shown in Figure 1b. XRD results show that the crystal structures of DOX-Fe_3_O_4_@agar at 2θ were 30.3°, 35.7°, 43.5°, 57.4°, and 62.9°, which were assigned to magnetite crystal planes (220), (311), (400), (511), and (440) (refer to JCPDS no. 85-1436). Therefore, biocompatible agar, as a drug carrier, has been confirmed to have the ability to encapsulate Fe_3_O_4_ magnetic particles. Simultaneously, suggesting that DOX exists in the amorphous form in DOX-agar and DOX-Fe_3_O_4_@agar [25,26,27,28,29]. The Fe_3_O_4_ particle size calculated by Scherrer’s formula was 9.2 nm, which was consistent with the particle size observed in TEM image. The SQUID was used to measure the magnetic field of DOX-Fe3O4@agar nanoparticles (NPs) (Figure 2), which obtained a saturation magnetization of 41.9 emu/g, showing superparamagnetism. This indicates that Fe_3_O_4_ in the core of the agar carrier still retained good magnetism after loading the drug.

### 3.2. Drug Release Capacity of DOX-Fe_3_O_4_@agar

According to literature, DOX has a stable UV absorption wavelength (around 480 nm) and red fluorescence emission of about 590 nm [30]. Therefore, the ability of DOX-Fe_3_O_4_@agar NPs to release the DOX drug was monitored by UV-vis spectrophotometer. In addition, based on the colonic pH value of 6.4–7.0 [31], DOX release effect was tested under physiological conditions of pH 7.0 in phosphate buffered saline (PBS) solution. As shown in Figure 3, after 56 min of treatment at pH 7, the release rate was 85% ± 3% (*n* = 3). According to the literature, agar gel is a material of open porous network system and spontaneous sustained release can be achieved. That have mesopores with an average pore radius of 4.7–8.6 nm for 4–6% (*v*/*v*) gel [32,33]. In addition, the DOX small-molecule is smaller than the porous network, which can migrate freely through these networks. This allows the drug release to reach balance within one hour. These results confirm the feasibility of the carrier itself in drug release.

Furthermore, DOX release from DOX-Fe_3_O_4_@agar nanoparticles was monitored thanks to DOX’s red fluorescence property, on HT-29 cells after 1 h treatment by fluorescence microscopy (Figure 4). The results showed that the HT 29 cells in the control group did not have fluorescence (Figure 4a,d). However, when cells were cultured with pure Dox for 1 h, the HT 29 cell image showed red fluorescence (Figure 4e). In addition, after culture for 1 h in the DOX-Fe_3_O_4_@agar treatment group (Figure 4c), red fluorescence could be clearly observed (Figure 4f), which demonstrates that DOX-Fe_3_O_4_@agar successfully releases DOX and can be applied to cells. This can be seen clearly with the merged images shown as the insert in Figure 4d–f.

### 3.3. DOX-Fe_3_O_4_@agar In Vitro Testing

To investigate the ability of DOX-Fe_3_O_4_@agar nanoparticle drug to inhibit the growth of cancer cells, we performed 3-(4,5-Dimethylthiazol-2-yl)-2,5-diphenyltetrazolium bromide (MTT) analysis to test the cell viability, as shown in Figure 5. The results showed that the longer the DOX-Fe_3_O_4_@agar nanoparticles were used to treat cells, the more significant was the decrease in cell viability (Figure 5a). On the other hand, it was also clearly observed under light microscope that, after drug treatment, HT-29 cells were significantly reduced compared with the control group (Figure 5b). The table in the lower part of Figure 5a shows that the cell viability of HT-29 cells decreased to 33% after 72 h of DOX treatment. However, DOX-Fe_3_O_4_@agar slowed down the rate of cell death, with a cell viability of 66%. We believe that the slow release of the encapsulated drug might reduce direct cell damage caused by excessive drug dose, or by diminishing drug accumulation.

### 3.4. Hyperthermia

In order to increase the functionality of the carrier, the ability of DOX-Fe_3_O_4_@agar NPs to generate heat in cell culture solution was evaluated (as shown in Figure 6). The results showed that the heating temperature of DOX-Fe_3_O_4_@agar increased with time. The heating temperature could reach 45 °C in 5 min, and the specific absorption rate (SAR) was 23.6 ± 1.1 W/g, showing good heating capacity. However, compared with pure Fe_3_O_4_ (SAR 46.8 ± 2.0 W/g), the heating capacity was slightly lower. The reason for this may be that the outer gel layer of Fe_3_O_4_ NPs absorbed water, causing the particles to swell, and increasing the dipole-dipole interaction between the particles, which reduces the rotational ability of particles and inhibits the Brownian heating mechanism. We further measured the intracellular heating capacity of DOX-Fe_3_O_4_@agar NPs. It was observed that the SAR value of heating capacity decreased to 18.9 ± 2.0 W/g, which might be due to the restriction of drug carrier particles by more space obstacles of cells. Even so, DOX-Fe_3_O_4_@agar NPs could still make the heating temperature reach about 43 °C in 5 min. As shown by previous studies, this temperature is sufficient to overcome the heat resistance for the synthesis temperature of cancer cell proteins, and thus exerts a cytotoxic effect by hyperthermia [34,35,36].

Finally, we added DOX-Fe_3_O_4_@agar NPs to HT-29 cells and cultured them for 24, 48, and 72 h, applied alternating magnetic field to induce hyperthermia for 15 min, and then carried out MTT test for cell viability, as shown in Figure 7. The results showed that the viability of HT-29 cells cultured with DOX-Fe_3_O_4_@agar was significantly reduced after hyperthermia. The viability of HT-29 cells decreased to 18.1% ± 10.7%, 16.9% ± 13.6%, and 15.4% ± 6.1% respectively after treatment for 24, 48, and 72 h.

## 4. Conclusions

Agar drug carriers showed a high release rate of 85% ± 3% under simulated colonic environment (pH 7). When HT-29 cells were cultured with the carrier particles, significant red fluorescence present in the cells could also be observed. In order to determine the magnetocaloric functionality, we evaluated the heating capacity of magnetic particles carrying drugs, which reached 43 °C after 5 min. This temperature is sufficient to exert a hyperthermic effect on cancer cells. Therefore, biocompatible agar, as a drug carrier, has been confirmed to have the ability to simultaneously encapsulate drugs and magnetic particles. This allows for the simultaneous application of chemotherapy and physical hyperthermia, and has a significant additive effect in killing cancer cells. Therefore, the commercial cost can be reduced, and the complexity of synthesis can also be reduced, thus presenting excellent potential application value in biomedical applications.

## Figures and Tables

**Figure 1 materials-14-01824-f001:**
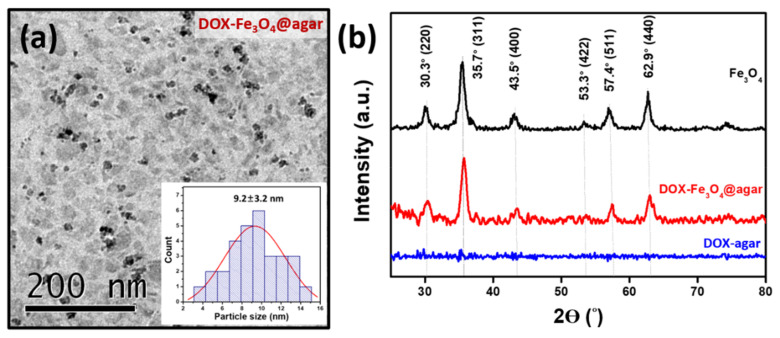
(**a**) TEM image and the inset shows the particle size distribution analysis by TEM image. (**b**) XRD spectrum of Fe_3_O_4_, DOX-agar and DOX-Fe_3_O_4_@agar.

**Figure 2 materials-14-01824-f002:**
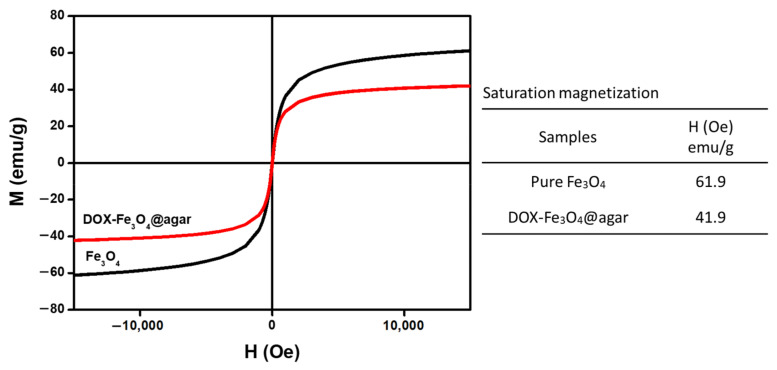
Magnetic strength of Fe_3_O_4_ and DOX-Fe_3_O_4_@agar as measured by SQUID magnetometer.

**Figure 3 materials-14-01824-f003:**
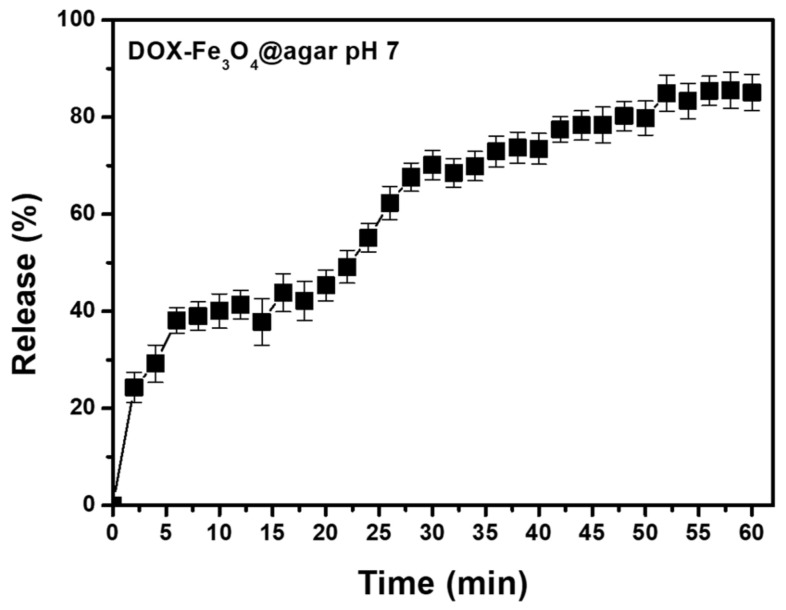
Drug release profiles of DOX-Fe_3_O_4_@agar at pH 7. These results shown are representative of at least three independent experiments.

**Figure 4 materials-14-01824-f004:**
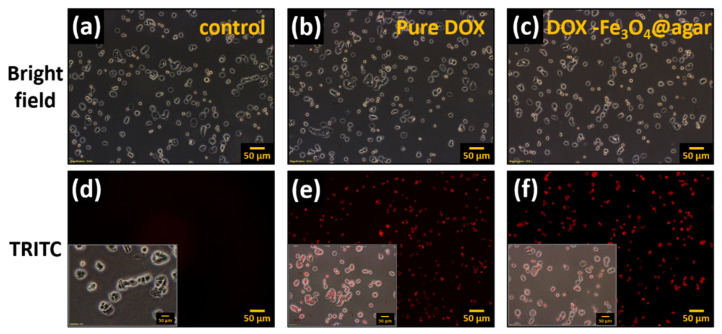
Fluorescence microscopy images of HT-29 cells after 1 h of incubation with pure DOX and DOX-Fe_3_O_4_@agar (on tetramethylrhodamine (TRITC) channel, at original magnification 100-fold); (**a**,**d**) control HT-29 cells, (**b**,**e**) pure DOX-treated cells, and (**c**,**f**) DOX-Fe_3_O_4_@agar-treated cells. Insert: Merged images of HT29 cells from bright and TRITC images.

**Figure 5 materials-14-01824-f005:**
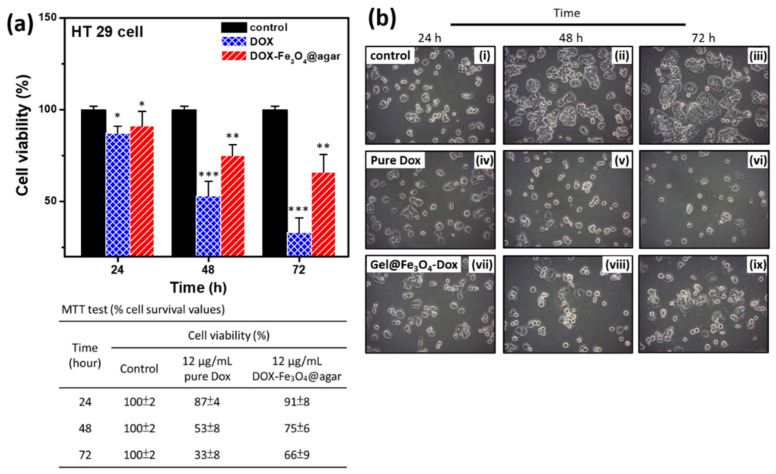
(**a**) Cell viability was monitored at multiple time points; 24 h, 48 h, and 72 h (*n* = 3). The detailed cell viability (%) values are shown in the table below. (**b**) Cell images were monitored at multiple time points; 24 h, 48 h, and 72 h. The images are control (i–iii), pure DOX-treated (iv–vi), and DOX-Fe_3_O_4_@agar-treated cells (vii–ix). Statistical analysis was performed using one-way ANOVA followed by Duncan’s test. * *p* < 0.05, ** *p* < 0.01, and *** *p* < 0.001 versus control.

**Figure 6 materials-14-01824-f006:**
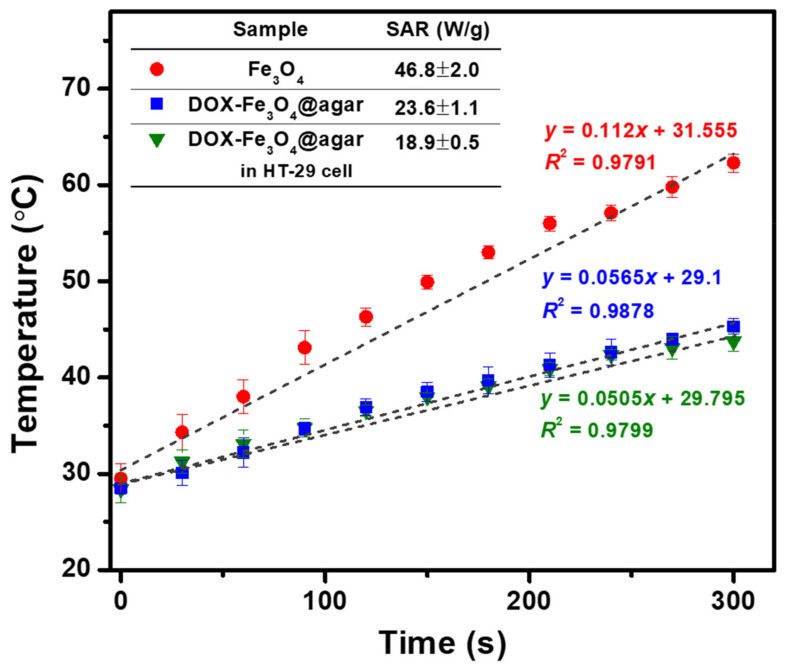
Temperature achieved by Fe_3_O_4_, DOX-Fe_3_O_4_@agar, and DOX-Fe_3_O_4_@agar in HT-29 cells. Here, 5 mg of sample is dispersed in 1 mL of DMEM (applied field = 400 A, f = 250 kHz). Inset shows the SAR values.

**Figure 7 materials-14-01824-f007:**
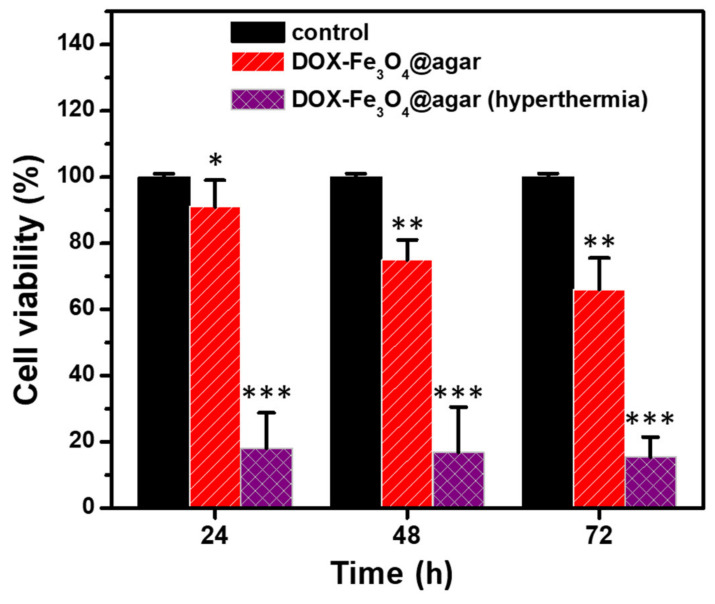
Cell survival of HT 29 cells treated with magnetic hyperthermia determined by MTT assay. The samples are control, pure DOX-treated (12 μg/mL), and DOX-Fe_3_O_4_@agar-treated (12 µg/mL) cells (*n* = 3). Statistical analysis was performed using one-way ANOVA followed by Duncan’s test. * *p* < 0.05, ** *p* < 0.01, and *** *p* < 0.001 versus control.

## Data Availability

No data.

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
