# Peer review of "Utilizing Edible Agar as a Carrier for Dual Functional Doxorubicin-Fe3O4 Nanotherapy Drugs"

_materials, 2021, doi:10.3390/ma14081824_

Round 1

Reviewer 1 Report

  1. TEM images show irregular spheroid shapes. It seems that there are so many aggregated particles.
  2. The XRD spectrum of DOX should be provided.
  3. How was the release of DOX decreased around 15 min? In addition, the release study should have been done for a prolonged period of more than 1 hour.
  4. The merged images should be provided in Figure 4.
  5. The authors should provide the statistical calculation in Figure 5(a) and 6.
  6. Scale bars are missing in Figure 5(b).

Reviewer 2 Report

The authors present their findings on the utilization of agar nanoparticles as a single vehicle for drug delivery as well as ferromagnetism. The manuscripts lacks in some critical aspects and the comments below should be addressed before it can be considered for publication:

  1. Title: Agar being mesoporous has been proven in other studies, but not in this study. Also, the use of the term "duel" brings in ambiguity as if drug release and ferromagnetism are competing phenomena. Hence, please consider changing the title so that it reflects the current study. Suggestion : "Utilizing edible agar as a carrier for dual functional doxorubicin-FE3O4 nanotherapy drugs".
  2. 2.1 Change the subtitle "Characterization" to "Materials".
  3. Section 2. The description of how the drug release study was done is missing. Please add this information.
  4. Section 2. Add the detailed description of TEM and XRD characterization methodologies.
  5. Figure 3. How many trails of drug release were done? Please include the mean values and corresponding standard deviations in the figure.

Round 2

Reviewer 1 Report

(Response 1) I didn’t ask the authors to provide references. Aggregated particles indicate instability during long-term storage as well as systemic circulation in plasma. 

(Response 2) How did DOX exhibit no peak in XRD? 

(Response 3) The authors described that the release study had done in 1ml of PBS with 5 mg of formulation. The volume was too little to conduct release studies. Furthermore, dialysis membrane or dialysis bag was not used even though nano-sized particles were tested. I strongly recommend that the authors should redo a release study using a proper method. 

Reviewer 2 Report

The authors have attended to the reviewers comments and have attended to the concerns appropriately. The manuscript is now in a much better shape and can be considered for publication. The manuscript will benefit from some further text editing, e.g. removing fullstop in line 177.
